# Genome-Wide Identification and Evolutionary Analysis of the GATA Transcription Factor Family in Nitrogen-Fixing Legumes

**DOI:** 10.3390/plants14162456

**Published:** 2025-08-08

**Authors:** Meng Xia, Liwen Tang, Haoming Zhai, Yezhou Liu, Liangsheng Zhang, Dan Chen

**Affiliations:** 1Hainan Institute, Zhejiang University, Sanya 572025, China; summer0501xm@outlook.com (M.X.); 22316216@zju.edu.cn (L.T.); 22316172@zju.edu.cn (H.Z.); 22316170@zju.edu.cn (Y.L.); 2Zhejiang Provincial Key Laboratory of Horticultural Plant Integrative Biology, College of Agriculture and Biotechnology, Zhejiang University, Hangzhou 310058, China; 3Yazhouwan National Laboratory, Sanya 572025, China

**Keywords:** GATA, legumes, symbiotic nitrogen fixation, phylogenetic analysis

## Abstract

GATA transcription factors are crucial for plant development and environmental responses, yet their roles in plant evolution and root nodule symbiosis are still not well understood. This study identified GATA genes across the genomes of 77 representative plant species, revealing that this gene family originated in Charophyta and significantly expanded in both gymnosperms and angiosperms. Phylogenetic analyses, along with examinations of conserved motifs and cis-regulatory elements in *Glycine max* and *Arabidopsis*, clearly demonstrated structural and functional divergence within the GATA family. Chromosomal mapping and synteny analysis indicated that GATA gene expansion in soybean primarily resulted from whole-genome duplication events. These genes also exhibit high conservation and signs of purifying selection in *Glycine max*, *Lotus japonicus*, and *Medicago truncatula*. Furthermore, by integrating phylogenetic and transcriptomic data from eight nitrogen-fixing legume species, several GATA genes were identified as strongly co-expressed with *NIN1*, suggesting their potential co-regulatory roles in nodule development and symbiosis. Collectively, this study offers a comprehensive overview of the evolutionary dynamics of the GATA gene family and highlights their potential involvement in root nodule symbiosis in legumes, thus providing a theoretical foundation for future mechanistic studies.

## 1. Introduction

GATA transcription factors are a crucial class of cis-regulatory elements prevalent in animals, plants, and fungi. Their defining characteristic is the ability to recognize and bind the conserved W-G-A-T-A-R DNA sequence, regulating diverse biological processes via a distinctive type IV zinc finger domain [1]. Compared to the relatively small and limited GATA gene repertoire in animals and yeast [2], the GATA gene family in plants has undergone substantial expansion. For instance, while the human genome contains only six GATA transcription factors, *Arabidopsis thaliana* has 30 identified GATA genes [3,4]. This marked expansion suggests a distinct evolutionary trajectory and functional diversification of GATA factors within the plant kingdom.

The plant GATA gene family is classified into four subfamilies (A, B, C, and D), a division strongly supported by phylogenetic analysis, conserved exon–intron organization, and shared motif architecture. These subfamilies are primarily distinguished by features within their single CX_2_CX_17–20_CX_2_C zinc finger domain. Subfamilies A and B both possess an 18-residue loop (CX_2_CX_18_CX_2_C). Subfamily A is defined by conserved Gln and Thr residues, while Subfamily B features a characteristic IRX(R/K)K motif in its C-terminal α-helix, suggesting distinct functionalities. Subfamily C contains a 20-residue loop (CX_2_CX_20_CX_2_C), hypothesized to have arisen from a two-amino-acid insertion. It also contains a Met-for-Trp substitution that may alter metal binding. Subfamily D is characterized by a highly conserved sequence, including a notable Gly-to-Glu substitution, implying specialized DNA-binding specificity [1,3,5].

Phylogenetic evidence indicates that plant GATA factors originated in early photosynthetic eukaryotes, such as green and red algae. The subsequent diversification into these distinct clades corresponds with the evolution of specialized regulatory roles in plant development and signal transduction [6,7]. Class A GATA factors are the largest subgroup in land plants, and these genes are frequently associated with photomorphogenesis and hormone responses [8]. For example, *AtGATA2* regulates root apical meristem and hypocotyl development via light and brassinosteroid signaling pathways [9,10], while *AtGATA12* is involved in xylem development and root growth [11]. Class B genes are divided into two subgroups characterized by the presence of either a HAN or LLM domain [12], and they primarily participate in chlorophyll biosynthesis, meristem maintenance, and general growth and development. Representative genes like *AtHAN* and *AtGNC* play critical roles in meristem boundary determination, floral organ formation, and nitrogen response [13,14]. Class C GATA factors are often associated with domains such as TIFY and CCT, suggesting potential roles in light signaling pathways and epigenetic regulation [15]. In contrast, Class D, the least abundant group, appears to be under strong selective pressure and possesses highly specific protein interaction capabilities [16].

Additionally, GATA factors are broadly involved in plant responses to environmental stresses, playing central roles in processes like light perception, nitrogen metabolism, metal ion transport, and carbon metabolism. For example, GATA genes regulate photoperiod adaptation, metal porphyrin biosynthesis, and circadian rhythm function, thereby indirectly influencing photosynthetic efficiency and nutrient assimilation capacity [17,18].

Nitrogen is an essential nutrient for plant growth, but its limited availability in natural soils often bottlenecks agricultural productivity [19]. Leguminous plants overcome this limitation through a unique nitrogen-fixing symbiosis system with rhizobia [20]. Globally, legume crops yield between 430 and 530 million tons annually, representing 8–10% of the total output of major crops. Soybean is the dominant food and economic legume, contributing more than 80% of the total legume yield and serving as a key source of protein and oil worldwide [21].

GATA transcription factors play key roles in nitrogen sensing and regulation. *In Saccharomyces cerevisiae*, *Gln3* and *Gat1* mediate nitrogen catabolite repression (NCR), while in *Yarrowia lipolytica*, *Gzf3* and *Gzf2* regulate ammonia utilization and lipid accumulation [22,23]. In higher plants, specific GATA factors directly modulate nitrogen use efficiency and assimilation; for example, *PdGNC* in poplar (*Populus clone 717*) influences nitrogen metabolism and biomass accumulation, while *CsGATA17* in tea plant (*Camellia sinensis*) correlates with nitrogen levels and nitrogenous metabolites like theanine and ethylamine [24,25]. Genome-wide analyses in soybean (*Glycine max*) further reveal significant up- or down-regulation of multiple GATA genes under low-nitrogen stress, suggesting their role in enhancing nitrogen limitation adaptation through metabolic pathway regulation [26]. Given the unique nitrogen-fixing capacity of legumes, a comprehensive analysis of the evolutionary patterns and functional diversification of GATA genes in these species may not only uncover their roles in regulating symbiosis but also provide novel insights into the identification of key functional genes.

Using the *Arabidopsis thaliana* GATA gene family classification system as a reference framework, we categorized GATA genes in target species into four subfamilies (A, B, C, and D) based on phylogenetic relationships, zinc finger domain characteristics, and gene structural conservation. *Arabidopsis*, as the earliest and most comprehensively annotated model for this family, provides a robust foundation for cross-species comparisons [3,5]. Leveraging this framework, we conducted a systematic comparative analysis and functional predictions of GATA genes across 77 plant species—including nitrogen-fixing legumes such as *Glycine max*, *Medicago truncatula*, and *Lotus japonicus*—to investigate their roles in nitrogen adaptation. Furthermore, we employed synteny and selection pressure analyses to investigate the conservation and functional divergence of GATA genes among different legume species. We also integrated transcriptomic data to identify GATA genes with organ-specific expression or a strong association with root nodule symbiosis. Ultimately, this study reveals the evolutionary dynamics of the GATA gene family in nitrogen-fixing legumes and offers new perspectives for understanding their potential roles in regulating symbiotic nitrogen fixation.

## 2. Results

### 2.1. Phylogenetic Diversification and Expansion of the GATA Gene Family During Plant Terrestrialization

To investigate the phylogenetic evolution of the GATA gene family, we comprehensively analyzed the genomes of 50 representative early land plants and algae, encompassing bryophytes, ferns, gymnosperms, and angiosperms. We identified a total of 921 GATA genes and classified them into four major classes—Classes A, B, C, and D—based on their conserved domain features and phylogenetic relationships (Figure 1; Appendix A).

The distribution of GATA genes varied significantly across evolutionary lineages. Statistical analysis revealed that Class A genes were entirely absent in algae, suggesting that this class likely emerged and expanded during land plant evolution. In contrast, Class B, C, and D genes were present across all major plant groups, indicating an early origin and strong evolutionary conservation. Charophyta, a group of green algae, represents the earliest lineage identified to harbor GATA genes, supporting the idea that this gene family played a pivotal role in the transition from aquatic to terrestrial plant life. This evolutionary expansion is especially evident in angiosperms, which possess a substantially higher number of GATA genes than earlier lineages. Notably, Class A and B genes underwent remarkable expansions in Nymphaeales and monocots, reaching 80 and 66 members, respectively.

We further analyzed the genomes of 19 gymnosperms and 5 angiosperms, identifying a total of 614 GATA genes (Figure 2; Appendix A). Our results show that the GATA gene family was already widespread in gymnosperms, indicating its origin and expansion predated the emergence of angiosperms. Welwitschia mirabilis was identified as the earliest gymnosperm representative harboring GATA genes, supporting its basal position within the gymnosperm lineage.

Within gymnosperms, the distribution of GATA gene classes varied: Class A genes constituted the largest proportion (46.05%), while Class D genes were the least represented (7.89%). This pattern suggests that Class A and B genes already had high expression potential and functional diversity in gymnosperms, whereas Class D genes may have expanded and diversified later during angiosperm evolution.

These findings further support the conserved yet expanding evolutionary trajectory of the GATA gene family in plants. Notably, the shift in the relative proportions of GATA gene classes from gymnosperms to angiosperms may reflect their functional diversification in response to light, hormonal regulation, and environmental adaptation.

### 2.2. Structural and Regulatory Diversification of GATA Transcription Factors in Soybean and Arabidopsis

To elucidate the evolutionary relationships and functional diversification of the GATA gene family, we performed a phylogenetic analysis on GATA genes from soybean (*Glycine max*) and *Arabidopsis thaliana* (Figure 3a; Appendix A). We identified a total of 61 GATA genes in soybean, comprising 28 Class A, 16 Class B, 9 Class C, and 8 Class D members. In *Arabidopsis*, 30 GATA genes were identified, with the majority belonging to Class A (14 genes), followed by Class B (11 genes), Class C (3 genes), and Class D (2 genes). The constructed phylogenetic tree supports the formation of well-defined clades during GATA gene evolution and demonstrates a high degree of phylogenetic conservation across different classes.

We performed a motif analysis of GATA proteins from soybean and *Arabidopsis* thaliana (Figure 3b), displaying 15 predicted conserved motifs as colored bars to show their distribution across different genes. Motif 1 was widely present in all species and across all classes of GATA proteins, suggesting its high conservation during the evolution. Class A members were generally enriched in motifs 2, 5, 6, and 11, and exhibited a greater diversity of motif types, indicating potential structural complexity and functional versatility. In contrast, Class B and Class C shared a relatively similar motif composition, while Class D had the simplest motif architecture, further implying a more conserved or simplified structural and functional role.

Domain analysis (Figure 3c) revealed that nearly all GATA proteins contain the characteristic GATA domain, underscoring its central role in DNA binding. Additionally, some Class A genes possess a C2H2-type zinc finger domain, suggesting they may have enhanced DNA-binding capacity or greater regulatory complexity. Other zinc finger types and auxiliary domains exhibited differentiated distributions among the various classes, reflecting the potential functional specificity of different GATA gene groups in distinct biological processes.

Further analysis of the 2000 bp upstream promoter regions of GATA genes revealed a wide distribution of cis-acting elements responsive to light, plant hormones, stress, and developmental signals in both *Arabidopsis thaliana* and *Glycine max* (Figure 3d). The CAAT-box and TATA-box were the most abundant elements in both species, with soybean generally showing higher counts. Most GATA genes contained light-responsive elements, including the G-box, Box-4, and GATA-motif. Furthermore, the presence of stress-responsive elements such as ARE, TC-rich repeats, WUN-motif, and LTR suggests GATA genes’ potential involvement in abiotic stress responses.

### 2.3. Genomic Landscape and Whole-Genome Duplication-Driven Expansion of GATA Genes in Glycine max

To clarify the genomic distribution and evolutionary relationships of GATA genes in soybean, we conducted a comprehensive analysis of their chromosomal localization and gene collinearity (Figure 4). All 61 GATA genes are distributed across all 20 soybean chromosomes (Chr1–Chr20), indicating broad genomic coverage of this gene family within the soybean genome.

Further classification of gene duplication types revealed that whole-genome duplication (WGD) or segmental duplication is the primary driving force behind the expansion of the GATA gene family. In addition, a number of genes were identified as dispersed or tandem duplicates, while singleton and proximal duplicates were relatively few. This suggests that large-scale duplication events have predominantly contributed to the amplification of GATA genes in soybean.

### 2.4. Evolutionary Dynamics of the GATA Gene Family in Legumes: Insights from Chromosomal Collinearity and Selection Pressure

To further investigate the evolutionary characteristics and conservation of the GATA gene family in legumes, we systematically analyzed the genomic distribution and collinearity relationships of GATA genes in *Glycine max*, *Medicago truncatula*, and *Lotus japonicus* (Figure 5). Our results showed that GATA genes are distributed across most chromosomes in all three species, exhibiting relatively dispersed arrangements with no significant clustering patterns. This widespread and scattered distribution may contribute to the diversification of gene regulatory functions.

Collinearity analysis revealed a high degree of conservation within the GATA gene family (Figure 5a). We identified numerous collinear GATA gene pairs among *Glycine max*, *Medicago truncatula*, and *Lotus japonicus*. This indicates that a large number of syntenic and conserved modules have been retained during the evolutionary history of legumes.

To further assess the selection pressure on the GATA gene family, we calculated the non-synonymous substitution rate (*Ka*), the synonymous substitution rate (*Ks*), and the *Ka/Ks* ratio for homologous GATA gene pairs among the three species. The *Ka/Ks* ratio is a key indicator of selective pressure on genes [27] (Figure 5b). Our results support that most GATA genes are under purifying selection (*Ka/Ks* < 1). This suggests these homologous genes have undergone evolutionarily conserved processes to maintain functional stability, with deleterious mutations being effectively eliminated. This trend implies that GATA genes may play central regulatory roles in legumes.

### 2.5. Organ-Specific Expression Profiling Reveals Functional Divergence of GATA Gene Subfamilies in Glycine max

To systematically reveal the tissue-specific expression patterns of the GATA gene family in soybean (*Glycine max*), we analyzed the expression profiles of 56 GATA genes across seven major tissue types: leaves, roots, root nodules, embryos, flowers, pods, and seeds. This analysis utilized data from the *Glycine max* RNA-seq database (https://plantrnadb.com/soybean/, accessed on 25 May 2025) [28]. To enhance the representativeness and clarity of the results, samples were carefully selected and categorized by tissue type. Subsequently, a heatmap of gene expression was generated using Z-score normalization for clearer visualization (Figure 6).

The heatmap analysis revealed that members of the soybean GATA gene family exhibit widespread yet distinct tissue-specific expression patterns. Specifically, Class A subfamily genes were predominantly expressed in reproductive organs, particularly in embryos, pods, and seeds. For example, *GLYMA16G155300* showed highly specific expression in flowers, and *GLYMA06G086400* was abundantly expressed in embryos, while *GLYMA07G108900* and *GLYMA04G008900* displayed strong expression in root nodules. In contrast, Class B members were mainly active in vegetative tissues, with several genes—such as *GLYMA13G103900*, *GLYMA17G055200*, *GLYMA14G094800*, and *GLYMA09G062800*—exhibiting high expression levels in leaves. Class C genes were broadly expressed in reproductive organs, including embryos, flowers, pods, and seeds. Class D genes showed more pronounced expression in vegetative tissues, particularly in leaves, root nodules, and roots.

These observed expression differences among tissues highlight the diverse roles of the GATA gene family in soybean, suggesting their involvement in regulating both plant development and organ-specific functions.

### 2.6. Phylogenetic Diversity and Co-Expression of GATA Subfamilies in Nitrogen Fixation

To gain deeper insights into the evolutionary relationships and functional diversification of the GATA gene family across various root-nodule nitrogen-fixing plants, we systematically identified GATA family members in eight representative species, yielding a total of 233 non-redundant GATA genes (Appendix A). The number of GATA genes varied among species. *Arachis hypogaea* and *Mimosa pudica* had relatively high numbers, with 40 and 44 members, respectively. This could be due to larger genome sizes or gene family expansion events. In contrast, species such as *Lotus japonicus* and *Datisca glomerata* contained only 17 and 18 GATA genes, respectively, suggesting the family may have undergone varying degrees of contraction across different species.

Subsequently, a phylogenetic tree was constructed based on the full-length amino acid sequences of the GATA proteins (Figure 7). The results clearly divided all genes into three major subfamilies: Class A, Class B, and Class C, consistent with previously established functional classifications of the GATA family. Further analysis revealed that Class A genes dominated in most species, with particularly high representation in *Lupinus albus* and *Arachis hypogaea*. This suggests that this subfamily may play a broader and more prominent functional role in nitrogen-fixing plants.

To further investigate the potential roles of GATA family genes in root nodule symbiosis, we conducted a co-expression analysis using *NODULE INCEPTION 1* (*NIN1*) as a key reference gene. Previous studies have established that the transcription factor encoded by *NIN1* is highly conserved in legumes and functions as a crucial downstream regulator in the LysM receptor kinase signaling pathway, directly participating in nodule organogenesis initiation and the regulation of nitrogen fixation-related genes [29]. For instance, *NIN1* regulates the expression of signaling factors such as *NSP1/NSP2*, *CYCLOPS*, and nitrogen-fixation genes like *nifHDK*, thereby influencing bacteroid formation and nodule development [30,31].

Based on a publicly available RNA-Seq dataset [32] (Libourel et al., DOI: 10.1038/s41477-023-01441-w), we calculated the Pearson correlation coefficients between GATA genes and *NIN1* in eight plant species (Appendix A). Genes exhibiting significant co-expression with *NIN1* (*p* < 0.05) were selected. A total of 36 GATA genes were found to be significantly correlated with *NIN1* expression. Among these, Class B members accounted for the largest proportion (17 genes), followed by Class A (14 genes), while only 5 genes belonged to Class C. These findings suggest that Class B and Class A GATA genes may play significant regulatory roles within the symbiotic signaling network of root nodule formation.

To further explore the potential roles of GATA family genes in root nodule symbiosis, a co-expression analysis was performed using *NSP1/NSP2* as key reference genes. *NSP1* and *NSP2* (Nodulation Signaling Pathway 1 and 2) are essential GRAS family transcription factors involved in legume-rhizobium symbiotic nitrogen fixation. They function by integrating calcium signaling to activate Nod factor signaling [33]. In *Medicago truncatula*, symbiotic signaling is completely abolished in *nsp1/nsp2* mutants, resulting in a failure to initiate nodule primordia [34]. The results revealed that in transcriptomes from eight nitrogen-fixing plant species, multiple GATA family members exhibited significant co-expression patterns with *NSP1/NSP2*, spanning the subfamilies A, B, and C.

### 2.7. Transcriptomic Analysis of GATA Genes: Expression Patterns and Co-Regulatory Networks in Nodule-Related Tissues

To further dissect the transcriptional regulation characteristics of GATA family genes during root nodule symbiosis, we systematically analyzed their expression patterns in root and nodule-related tissues across eight nitrogen-fixing plant species. This analysis was based on the transcriptomic dataset published by Libourel et al. (2023) in *Nature Plants* [32]. Additionally, we explored potential co-expression relationships with the key symbiotic regulator *NODULE INCEPTION 1* (*NIN1*). In total, expression data from 220 samples were collected and analyzed, covering several representative legume species including *Mimosa pudica, Medicago truncatula, Arachis hypogaea*, and *Aeschynomene evenia* (Figure 8).

A total of 44 GATA family members were identified in the *Mimosa pudica* genome (Figure 8a). Expression profiling revealed that many Class A and B genes exhibited detectable expression in nodule development-related samples. Notably, genes such as *Mimpud_GATA11, Mimpud_GATA10*, and *Mimpud_GATA39* showed a significant positive correlation with *NIN1* expression (*p* < 0.05), suggesting their potential involvement in the regulatory network of nodule organogenesis.

In *Medicago truncatula*, 28 GATA family genes were identified (Figure 8b), comprising 18 from Class A, 7 from Class B, and 3 from Class C. Several Class A genes (e.g., *Medtru_GATA14, Medtru_GATA9,* and *Medtru_GATA6*) displayed high expression levels during key stages of nodule development, indicating their possible roles in nodule induction.

A total of 40 GATA genes were identified in *Arachis hypogaea* (Figure 8c). Expression analysis showed that several Class B genes (such as *Arahyp_GATA31*, *Arahyp_GATA24*, and *Arahyp_GATA29)* were highly expressed in nodule tissues and exhibited significant correlation with *NIN1,* implying their potential involvement in the formation and development of peanut’s typical nodules.

As an atypical nitrogen-fixing legume, *Aeschynomene evenia* harbors 20 GATA genes, with Class A and B members each accounting for half (Figure 8d). Notably, *Aeseve_GATA6* (Class A) and *Aeseve_GATA13* (Class B) were significantly expressed in nodule tissues and strongly co-expressed with *NIN1*, suggesting their involvement in regulating the development of the species’ unique “crack-entry” type nodules [35].

Expression profiles of GATA genes across different species demonstrate strong clade specificity and functional diversity. Notably, Class A and Class B members are frequently and highly expressed in nodule-related tissues, exhibiting significant positive correlations with the key symbiotic regulatory factors *NIN1* and *NSP1/NSP2*. These findings suggest that the GATA gene family may be involved in regulating nodule organogenesis and symbiotic signaling pathways in leguminous plants.

## 3. Discussion

In this study, members of the GATA gene family were comprehensively identified across 77 representative plant species, ranging from lower to higher plants, using systematic bioinformatics approaches. This provides an evolutionary perspective for uncovering the origin, expansion, and conservation of the GATA gene family.

Our phylogenetic analysis reveals an ancient origin for the GATA transcription factor family, with its earliest members traceable to aquatic charophytes. Subfamilies B, C, and D are highly conserved across major plant lineages, implying deep functional conservation. In contrast, Class A genes, which are absent in algae, expanded significantly in terrestrial plants, suggesting a key role in land adaptation through enhanced regulation of nutrient uptake and developmental plasticity. A notable expansion also occurred in gymnosperms, increasing regulatory network complexity before the emergence of angiosperms. This evolutionary trajectory continued in angiosperms, where lineage-specific selection and whole-genome duplication events drove further diversification, particularly in legumes. For instance, Class D genes, which diversified in angiosperms, exhibit strong purifying selection and predominant root expression, suggesting specialized roles in root development. This expansion likely provided the genetic redundancy that enabled legumes to fine-tune nitrogen metabolism and evolve specialized functions for symbiotic nitrogen fixation.

At the levels of gene structure and cis-regulatory elements, GATA genes generally possess the characteristic Cys_2_–Cys_2_-type zinc finger domain, with highly conserved gene structures within each subfamily [36]. Promoter cis-element analysis revealed that these GATA genes are enriched with hormone- and environment-responsive elements, including regulatory motifs related to light, various hormones (such as auxin, gibberellin, ABA, and MeJA), and diverse stress signals (e.g., cold and drought) [37,38,39]. For instance, Kim, M. et al. predicted light-, hormone-, and stress-responsive elements in the GATA gene family of populus [40]. Similarly, GATA promoters in wheat are rich in elements like ABA-responsive motifs, G-box (light-responsive), P-box (gibberellin-responsive), and AuxRE (auxin-responsive) [41]. These structural features suggest that GATA transcription factors may mediate plant responses to environmental cues and regulate nodule development and nitrogen metabolism via hormone signaling pathways. However, experimental validation of these predicted cis-elements and their specific regulatory functions is essential to clarify how GATA-mediated networks integrate multiple signals to control nodulation and nitrogen use.

In soybean, GATA genes are distributed across multiple chromosomes, with numerous gene pairs indicating prevalent segmental duplication. Similarly, Ka/Ks ratios <1 for wheat GATA genes suggest strong purifying selection [41], consistent with evolutionary conservation observed in legumes.

Expression pattern analysis revealed that legume GATA genes exhibit significant differential expression across various tissues. In soybean, GATA gene expression demonstrates clear tissue specificity: Some GATA genes are highly expressed in leaves or stems, while others are more enriched in roots. It is known that overexpression of *GmGATA44* can compensate for the reduced chlorophyll content phenotype in *Arabidopsis AtGATA21 (GNC)* mutants, indicating that *GmGATA44* plays an important role in regulating chlorophyll biosynthesis and nitrogen assimilation [26,42]. These findings align with previous reports of GATA family involvement in nitrogen metabolism and photosynthetic regulation [6,8].

It is particularly noteworthy that we identified several GATA genes in various legume species exhibiting strong co-expression with *NIN1*, a key nodulation regulatory factor. Previous studies have also shown that GATA transcription factors negatively regulate nodule formation in model legume species. For instance, in *Medicago truncatula*, simultaneous mutation of two GATA genes homologous to *Arabidopsis HANABA TARANU* (*MtHAN1* and *MtHAN2*) resulted in twice as many nodules as wild-type plants, revealing their inhibitory role in nodule differentiation [43]. This functional feature aligns with our observations: Certain GATA genes are highly expressed in nodule-associated tissues, and their promoters contain cis-acting elements related to hormones such as cytokinin and gibberellin, suggesting they might be regulated by nitrogen-fixation signaling and influence nodule development.

This study presents a genome-wide characterization of the GATA gene family in nitrogen-fixing legumes, defining their evolutionary relationships and structural features. Integrated transcriptomic and co-expression analyses identified candidate genes with potential roles in nodulation and nitrogen fixation. These bioinformatic predictions, while providing valuable hypotheses, necessitate empirical validation. Future functional characterization using techniques such as targeted gene knockout, overexpression, and promoter–reporter assays is crucial for confirming the precise roles of these genes.

Furthermore, our analysis of cis-regulatory elements suggests their involvement in coordinating nitrogen-responsive transcription. The accuracy of these predictions is inherently limited by the variable quality and completeness of current genome annotations. Therefore, improved genome assemblies will be critical for refining future comparative and regulatory studies.

A detailed understanding of how GATA transcription factors regulate symbiotic nitrogen fixation could provide powerful tools for molecular breeding. These insights are critical for developing crops with improved nitrogen use efficiency, offering a promising path toward sustainable agriculture and reduced dependence on synthetic fertilizers.

## 4. Materials and Methods

### 4.1. Identification and Phylogenetic Analysis of the GATA Gene Family

To identify members of the GATA transcription factor family, data from 77 representative plant species, comprising 19 gymnosperms, 31 angiosperms, and 27 lower plants, were integrated. Protein sequences, coding sequences (CDS), and genome annotations (GFF) were sourced from multiple public databases: NCBI (https://www.ncbi.nlm.nih.gov/, accessed on 10 June 2025), Phytozome v13 (https://phytozome-next.jgi.doe.gov/, accessed on 10 June 2025), Ensembl Plants (https://plants.ensembl.org/index.html, accessed on 10 June 2025), the National Genomics Data Center (https://download.cncb.ac.cn/gwh/Plants/, accessed on 10 June 2025), and Soybase (https://www.soybase.org/, accessed on 10 June 2025) [44,45,46,47,48].

First, the GATA zinc finger domain HMM profile (PF00320) [49] was used to scan all proteomes using HMMER v3.3.2 with permissive thresholds (E-value ≤ 1.0; inclusion cutoff = 0.01) to minimize false negatives [50]. Subsequently, candidate sequences were filtered to remove redundant entries and pseudogenes via CD-HIT (identity > 90%) and manual curation [51]. The presence of complete GATA conserved domains in the remaining sequences was then verified using both SMART (http://smart.embl-heidelberg.de/, accessed on 5 June 2025) and the NCBI Conserved Domain Database (CDD) [52,53]. See Appendix A for detailed information on the GATA genes identified across diverse plant lineages.

For the phylogenetic analysis, the confirmed GATA protein sequences were aligned using MAFFT v7.475 [54], and regions of poor alignment were manually trimmed in MEGA7 [55]. A maximum likelihood (ML) phylogenetic tree was then constructed using IQ-TREE v2.2.0 [56]. The optimal substitution model was determined based on the Bayesian Information Criterion (BIC), and branch reliability was assessed with 1000 bootstrap replicates. Finally, the resulting phylogenetic tree was visualized and annotated employing the online tool tvBOT (https://www.chiplot.online/tvbot.html, accessed on 5 June 2025) [57].

### 4.2. Motif Analysis, Domain Annotation, and Promoter Analysis of GATA Genes

#### 4.2.1. Conserved Motif Analysis

Conserved motifs within the GATA protein sequences were identified using MEME Suite v5.4.1. The analysis was executed with the mode set to ‘anr’ (any number of repetitions). The parameters were configured to detect a maximum of 15 motifs, with the width of each motif restricted to a range of 10 to 200 amino acid residues [58]. Visualization of the analysis was conducted using R (version 4.3.2).

#### 4.2.2. Domain Annotation

Protein domains were annotated using the ‘pfam_scan.pl’ script against the Pfam database (Pfam-A, version 35.0). The scan was executed with an E-value threshold set to 10^−5^. The presence of the typical GATA-type zinc finger domain was confirmed by cross-referencing the conserved domain information obtained from sequence alignments. Sequences that contained incomplete domains or were identified as potential false positives were subsequently excluded from the analysis [50]. The domain architecture of the confirmed proteins was visualized using R.

#### 4.2.3. Promoter Sequence Extraction and Cis-Acting Element Analysis

For each GATA gene, the 2000 bp region upstream of the start codon was extracted from the corresponding reference genome to serve as the promoter sequence, a task accomplished using Seqkit v0.16.1 [59]. The extracted promoter sequences were then analyzed with the PlantCARE database (http://bioinformatics.psb.ugent.be/webtools/plantcare/html/, accessed on 5 June 2025) to predict the locations of cis-acting regulatory elements [60]. Based on the output from PlantCARE, the identified elements were functionally categorized. Only those functional elements that occurred in at least five different gene promoters (occurrence ≥ 5) were retained for analysis. These filtered elements were subsequently classified into four functional categories: light responsiveness, plant hormone related, stress related, and plant growth and development. Visualization was conducted using R.

### 4.3. Chromosomal Localization and Evolutionary Analysis of GATA Genes

The chromosomal localization, collinearity, and evolutionary pressures on GATA genes were analyzed in *Glycine max*, *Medicago truncatula*, and *Lotus japonicus*. For intra-species analysis, an all-versus-all BLASTP alignment of GATA protein sequences was first performed (E-value threshold: 10^−10^; max hits: 5) [61]. Collinear gene pairs were subsequently identified from these alignment data using MCScanX [62]. The ‘duplicate_gene_classifier’ utility was then employed to categorize gene duplication events into five types: singleton, dispersed, proximal, tandem, and segmental/WGD. Chromosomal locations and intra-species collinearity were visualized using a Circos plot generated with the R package circlize [63].

Cross-species collinearity was examined using the Python JCVI toolkit [64]. Sequence databases were constructed with ‘lastdb,’ and alignments were performed using ‘lastal’ to identify syntenic anchor files. Subsequently, macro-synteny maps illustrating the conserved genomic regions among the three legume species were generated with the ‘jcvi.graphics.karyotype’ module.

To assess the selective pressures on duplicated genes, the ratio of non-synonymous (*Ka*) to synonymous (*Ks*) substitution rates was calculated for all identified collinear gene pairs. The *Ka*/*Ks* values were computed using KaKs_Calculator 2.0 with the Nei-Gojobori (NG) model [65]. *Ka*/*Ks* ratio distributions were visualized as box plots in R to compare selective pressures across species and collinear groups.

### 4.4. Expression Profile Analysis of GATA Genes in Different Soybean Tissues

The expression profiles of 56 soybean GATA genes were analyzed using RNA-Seq data procured from the *Glycine max* RNA-seq database [28] (https://plantrnadb.com/soybean/, accessed on 25 May 2025). This resource provided normalized expression data derived from 4085 transcriptome samples, which were sourced from seven major tissue types: leaf (n = 995), root (n = 856), nodule (n = 35), embryo (n = 138), flower (n = 44), pod (n = 39), and seed (n = 644). Detailed metadata, including sample titles, tissue types, cultivars, genotypes, treatments, accession numbers, sequencing depth, and mapping statistics, are provided in Appendix A.

The expression matrix was processed in the R environment (v4.3.1). To create a balanced dataset for visualization, a representative subset of samples was curated. For tissues with high sample counts (leaf, root, and seed), approximately 40 samples were randomly selected. For tissues with fewer available samples (nodule, flower, pod, and embryo), all samples were retained. The seven tissue types were further annotated and grouped into two broader developmental categories: vegetative organs and reproductive organs. The expression values for each gene across the curated sample set were then normalized using a Z-score transformation. Finally, a heatmap was generated using the ‘ComplexHeatmap’ R package (v2.14.0) to visualize the expression patterns [66].

### 4.5. Expression Analysis of GATA Genes During Root Nodule Symbiosis

The expression dynamics of GATA genes during symbiotic nitrogen fixation were investigated using a public RNA-Seq dataset published by Libourel et al. (2023) [32]. This dataset comprised 220 samples from eight nitrogen-fixing species, detailing gene expression across various symbiotic stages. The analyzed species and sample numbers included *Mimosa pudica* (n = 41), *Datisca glomerata* (n = 12), *Lupinus albus* (n = 6), *Aeschynomene evenia* (n = 18), *Arachis hypogaea* (n = 33), *Lotus japonicus* (n = 24), *Medicago truncatula* (n = 79), and *Hippophae rhamnoides* (n = 6). Detailed information on species, sample names, inoculated rhizobial strains, time points, developmental stages, number of biological replicates, SRA accession codes, and corresponding references is provided in Appendix A.

From the provided expression matrices, GATA gene identifiers were located by referencing the genome annotation file for each respective species. Expression levels for these genes, reported as fragments per kilobase of transcript per million mapped reads (FPKM), were extracted from all root and nodule-related samples. To identify GATA genes potentially involved in nodulation, a co-expression analysis was performed using the key marker gene *NODULE INCEPTION 1 (NIN1)* as a reference [29,30,31]. The Pearson correlation coefficient between the expression profile of each GATA gene and *NIN1* was calculated in R (v4.3.1) [67]. A relationship was considered statistically significant if the *p*-value was less than 0.05. The results of the co-expression analysis, including correlation coefficients and expression levels across tissues, were visualized as circular heatmaps. These figures were generated in R using the ‘circlize’ (v0.4.15) and ‘ComplexHeatmap’ (v2.12.1) packages [63,66].

## 5. Conclusions

This study provides a genome-wide analysis of the GATA transcription factor family across 77 plant species, tracing its ancient origins to charophytes and its subsequent expansion and diversification throughout land plant evolution. Our analyses confirm the deep conservation of Classes B, C, and D, while Class A likely acquired novel functions facilitating terrestrial adaptation. In legumes, this family’s expansion was driven by segmental and whole-genome duplications, with resulting paralogs maintained under strong purifying selection. Furthermore, transcriptome and co-expression network analyses identified candidate GATA genes implicated in nodulation, evidenced by strong co-expression with the master regulator *NIN1* and links to hormone signaling pathways. Ultimately, this research provides key insights and promising genetic targets for breeding crops with enhanced nitrogen use efficiency, thereby advancing sustainable agriculture.

## Figures and Tables

**Figure 1 plants-14-02456-f001:**
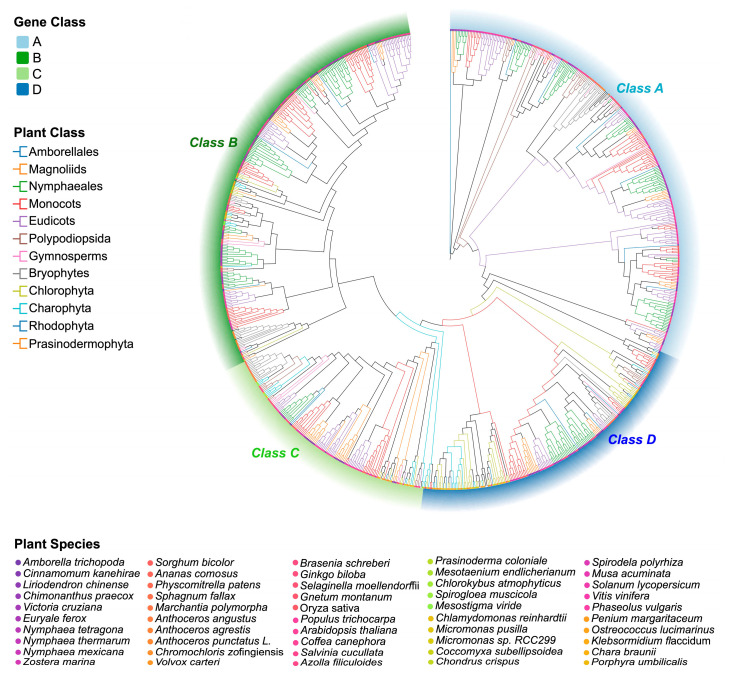
Phylogenetic analysis of GATA genes from 50 early land plant and algal species. The maximum likelihood (ML) phylogenetic tree displays four major classes, indicated by colored blocks: Class A (light blue), Class B (dark green), Class C (light green), and Class D (dark blue). Branch colors denote major taxonomic groups, while symbols at the branch tips represent individual species (full list provided in the legend). The tree was constructed with IQ-TREE v2.2.0 using the best-fit substitution model selected by the Bayesian Information Criterion (BIC). Branch support values were derived from 1000 bootstrap replicates.

**Figure 2 plants-14-02456-f002:**
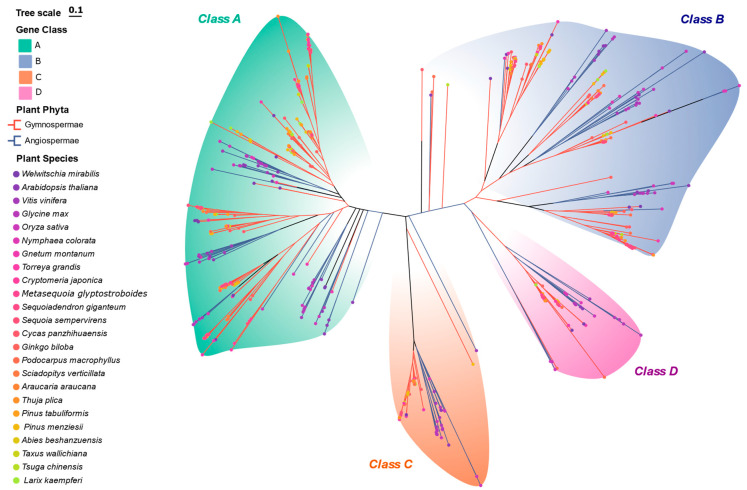
Phylogenetic relationships of GATA genes in gymnosperms and angiosperms. The ML phylogenetic tree of GATA genes from 19 gymnosperm and 5 angiosperm species. Genes were classified into four classes based on conserved domain characteristics, indicated by colored blocks: Class A (green), Class B (blue), Class C (orange), and Class D (pink). Branch colors differentiate angiosperms from gymnosperms. The tree was generated with IQ-TREE v2.2.0, and the optimal substitution model was selected via the BIC. Nodal support was assessed with 1000 bootstrap replicates. The scale bar indicates substitutions per site.

**Figure 3 plants-14-02456-f003:**
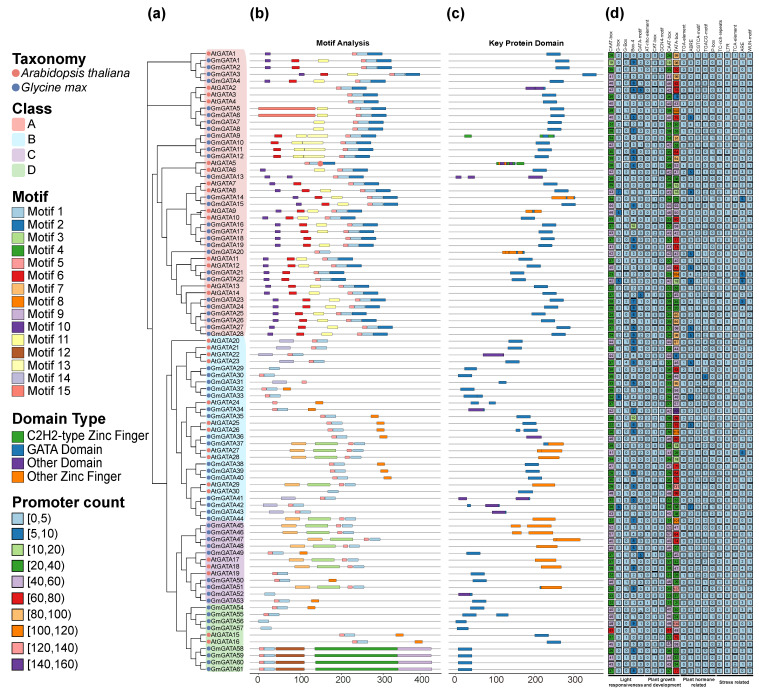
Structural and regulatory features of GATA genes in *Arabidopsis thaliana* and *Glycine max.* (**a**) The ML phylogenetic tree of GATA proteins. The four major GATA classes are denoted by distinct branch colors: Class A (red), Class B (blue), Class C (purple), and Class D (green). The tree was constructed using IQ-TREE v2.2.0 with nodal support from 1000 bootstrap replicates. (**b**) Conserved protein motifs. Motifs were identified using MEME Suite v5.4.1 (15 motifs maximum). Each colored box represents a distinct motif, with their arrangement shown relative to protein length. (**c**) Protein domain architecture. Domains were identified from the Pfam-A v35.0 database. Colored bars correspond to specific domain types: C2H2-type zinc finger (green), GATA domain (blue), other zinc finger domains (orange), and other domain types (purple). (**d**) Cis-regulatory elements in promoter regions. These elements were classified into four functional categories: light-responsive, hormone-responsive, stress-responsive, and developmental elements. The color intensity of each box represents the number of elements within the corresponding category.

**Figure 4 plants-14-02456-f004:**
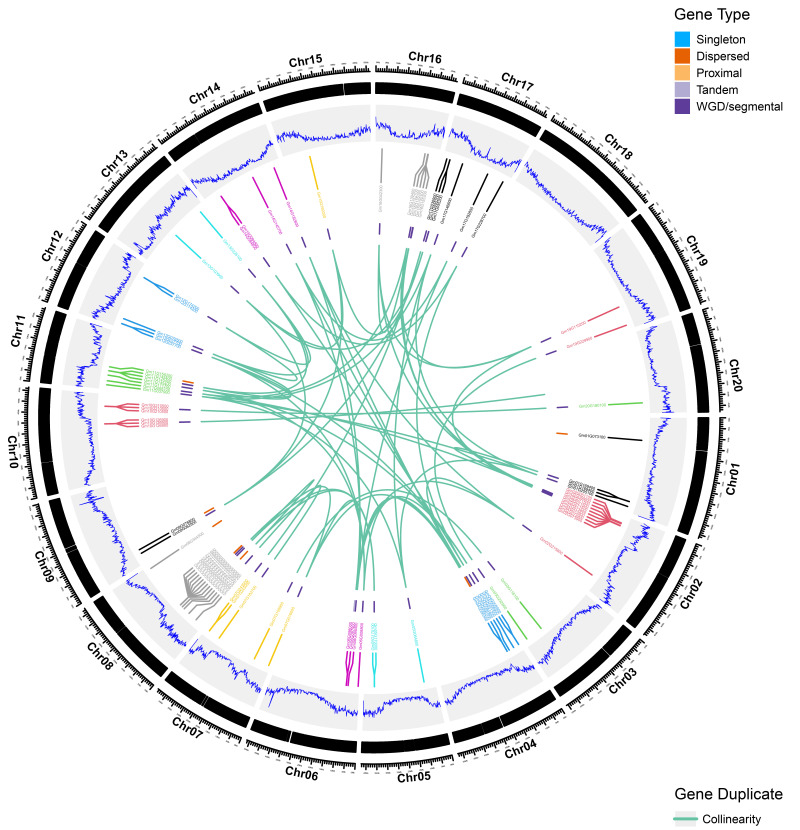
Chromosomal distribution and synteny of GATA genes in *Glycine max*. A Circos plot displaying genomic features of the soybean GATA gene family. Tracks are arranged from outermost to innermost: ideograms of the 20 soybean chromosomes (Chr01–Chr20); chromosomal coordinates in megabases (Mb); physical locations of GATA genes, marked by black rectangles; GATA gene density (blue line) calculated in a 500 kb sliding window; GATA gene identifiers, colored by chromosome; gene duplication event types: singleton (blue), dispersed (orange), proximal (yellow), tandem (light purple), and segmental/WGD (dark purple); and syntenic relationships between GATA genes, indicated by green lines. Duplication and synteny analyses were performed with MCScanX based on the *Glycine max* Wm82.a2.v1 reference genome.

**Figure 5 plants-14-02456-f005:**
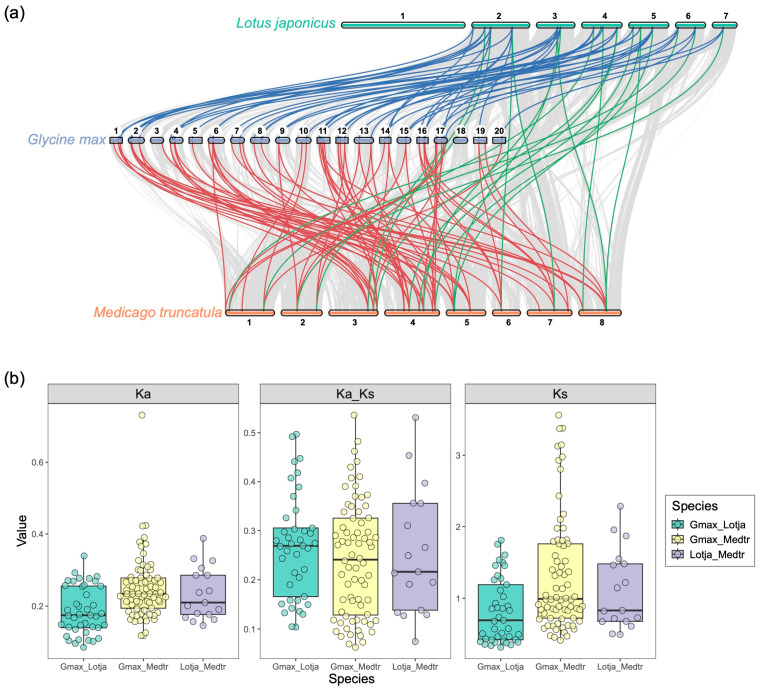
Interspecies synteny and selection pressure on GATA genes in three legume species. (**a**) Syntenic relationships of GATA genes between *Glycine max* (purple), *Lotus japonicus* (green), and *Medicago truncatula* (orange). Colored lines connect syntenic GATA gene pairs between species, while gray lines represent synteny of all other collinear genes. (**b**) Distribution of nonsynonymous substitution rates (*Ka*), synonymous substitution rates (*Ks*), and their ratios (*Ka/Ks*) for syntenic GATA gene pairs (*G. max*–*L. japonicus*, *G. max*–*M. truncatula*, and *L. japonicus*–*M. truncatula*).

**Figure 6 plants-14-02456-f006:**
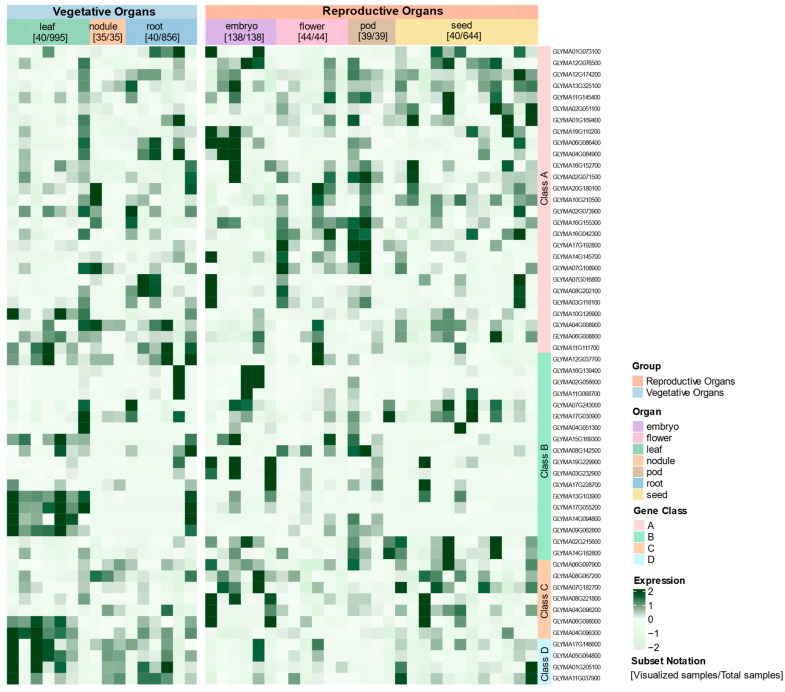
Expression profiles of *Glycine max* GATA genes across various tissues. Heatmap shows Z-score normalized expression of 56 genes (rows, grouped phylogenetically A–D) in 196 representative samples (columns). Samples represent seven organs stratified by tissue type: vegetative (leaf [40/995], root [40/856], and nodule [35/35]) and reproductive (embryo [138/138], flower [44/44], pod [39/39], and seed [40/644]), with full representation for smaller tissues and random subsets for high-volume tissues. Color gradient (light to dark green) indicates low to high expression. Data from *G. max* RNA-seq database [28].

**Figure 7 plants-14-02456-f007:**
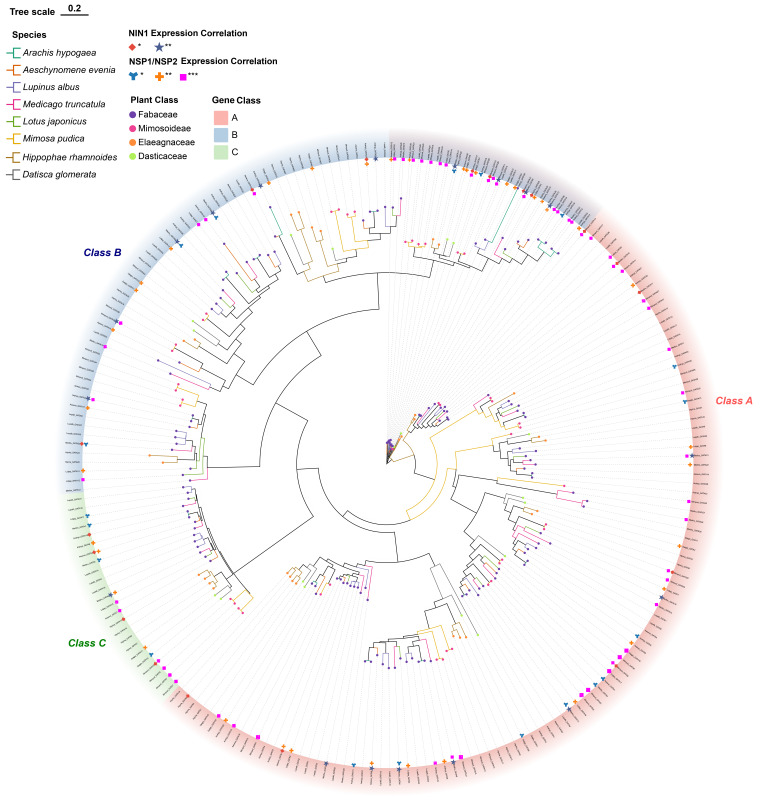
Phylogeny and expression correlation of GATA genes with *NIN1* and *NSP1/NSP2* in eight nitrogen-fixing species. Maximum-likelihood phylogeny of GATA proteins from *Arachis hypogaea*, *Aeschynomene evenia*, *Lupinus albus*, *Medicago truncatula*, *Lotus japonicus*, *Mimosa pudica*, *Hippophae rhamnoides*, and *Datisca glomerata*. Tree constructed with IQ-TREE using BIC-selected model (1000 bootstrap replicates; scale bar = 0.2). Genes classified into Class A (red), B (blue), or C (green) based on conserved domains. Branches are colored by species; symbols denote taxonomic groups. Pearson correlations between GATA genes and the expression levels of *NIN1* and *NSP1/NSP2* are indicated as follows: * *p* < 0.05, ** *p* < 0.01, *** *p* < 0.001.

**Figure 8 plants-14-02456-f008:**
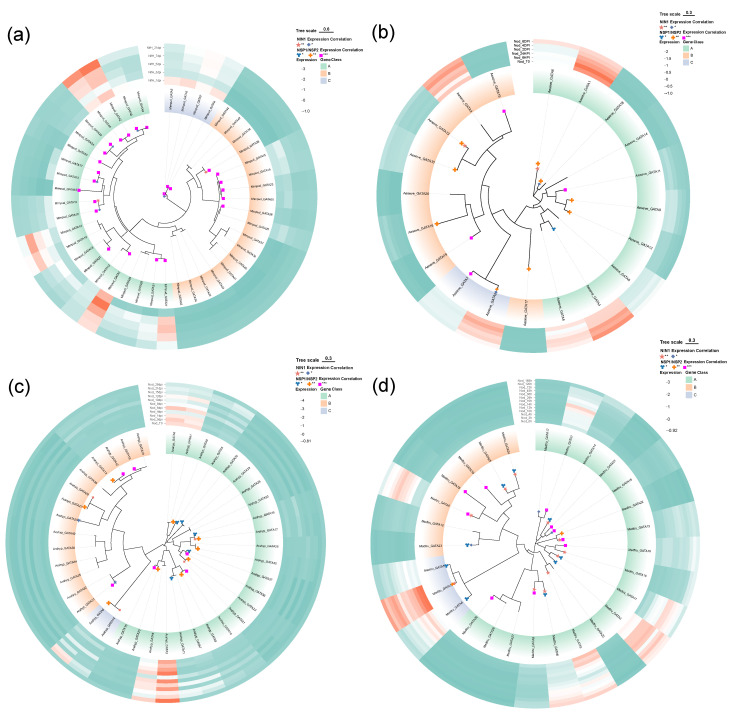
Phylogeny and expression dynamics of GATA genes during nodulation in four species. Phylogenetic trees (IQ-TREE, ML, BIC model, 1000 bootstraps; scales: *M. pudica* 0.6, others 0.3) with GATA classes: A (green), B (orange), and C (purple). Circular heatmaps: Z-score normalized expression (green-low to red-high). Pearson correlations between GATA genes and the expression levels of *NIN1* and *NSP1/NSP2* are indicated as follows: * *p* < 0.05, ** *p* < 0.01, *** *p* < 0.001. (**a**) *M. pudica* inoculated with *C. taiwanensis LMG19424* (nifH); samples were collected at 1, 3, 5, 7, and 21 days post-inoculation (NifH_1dpi–21dpi). (**b**) *A. evenia* inoculated with *Bradyrhizobium* sp. *ORS278*; samples were collected at 0 h (Nod_T0) and from 6 h to 6 days post-inoculation (Nod_6HPI–6DPI). (**c**) *A. hypogaea* inoculated with *Bradyrhizobium* sp. *SEMIA6144*; samples were collected from 0 to 28 days post-inoculation (Nod_T0–28dpi). (**d**) *M. truncatula* inoculated with *Sinorhizobium meliloti*; samples were collected from 0 h to 168 h post-inoculation (Nod_0h–168h).

## Data Availability

Data are contained within the article and Appendix A.

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
