# Peer review of "Genome-Wide Identification and Evolutionary Analysis of the GATA Transcription Factor Family in Nitrogen-Fixing Legumes"

_plants, 2025, doi:10.3390/plants14162456_

Round 1

Reviewer 1 Report

Comments and Suggestions for Authors

The aim of the study was to perform the genome-wide identification and evolutionary analysis of the GATA transcription factor family in nitrogen-fixing legumes. The study reveals that the GATA transcription factor family originated in Charophyta and underwent substantial expansion in seed plants, particularly in soybean through whole-genome duplications, accompanied by clear structural and functional diversification.

The results presented in the manuscript are interesting, valuable, and contribute meaningfully to the understanding of GATA transcription factors in plant evolution and root nodule symbiosis. However, the manuscript would benefit from the inclusion of a dedicated Conclusions section to clearly summarize the key findings and their implications.

Additionally, the Discussion section requires further refinement and significant expansion. In particular, it should address the functional limitations of bioinformatic predictions, the need for experimental validation, and potential evolutionary implications of GATA gene expansion. Including more considerations related to cis-regulatory element relevance, variability in genome annotation quality, and the translational potential of the findings would greatly enhance the overall impact and clarity of the manuscript.

The font size used in the figure labels and axis annotations should be increased to improve readability and ensure that all text is clearly visible.

The statistical analyses should be described in greater detail.

Author Response

We sincerely thank the reviewers for their constructive and insightful comments. We have carefully revised the manuscript in response to each suggestion and provided a detailed point-by-point response in the attached file.

Reviewer 2 Report

Comments and Suggestions for Authors

Major concerns

  1. What are the standards for the four major classes in Figure 1? It is not obvious enough to see the differences according to the phylogenetic tree.
  2. Are the several genes between class C and D belongs to class B in Figure 2? If so, why they are evolutionarily close to C and D?
  3. The sample numbers should be labeled in Figure 6.
  4. Are there other genes containing similar functions with NIN1? If so, these genes should also be used for co-expression analysis.
  5. The authors should be more cautious about gene functions in discussion part since there are no direct experimental proofs.

Author Response

(The authors gave the same response as above.)

Reviewer 3 Report

Comments and Suggestions for Authors

The manuscript presents a genome-wide identification of the GATA transcription factor family and conducts evolutionary and comparative analyses across key legume species, including soybean (Glycine max). It specifically focuses on analyzing gene expression patterns in the context of symbiotic nitrogen fixation. The overall logic of the manuscript is coherent, and the writing is generally clear. While the primary novelty lies in exploring the potential role of GATA transcription factors in legume nitrogen fixation, the conclusions are solely based on re-analyzed transcriptomic data without any experimental validation. Crucially, even basic expression validation (e.g., qRT-PCR) is absent. Consequently, the proposed involvement of specific GATA transcription factors in nitrogen fixation requires further experimental validation and should be interpreted with caution.

Major Concerns:

To address concerns about completeness of your methodology employed for identifying homologous genes across 77 genomes, clear description for the designing of the maximize detection sensitivity should be added to minimize the risk of omissions.

The entire study relies on in silico transcriptomic analyses of existing datasets. No wet-lab experiments (e.g., gene expression validation under nitrogen-fixing conditions via qRT-PCR or phenotypic analysis of mutants) were performed to substantiate the proposed roles of GATA factors in nitrogen fixation. This significantly weakens the biological significance and conclusions of the study. Robust experimental evidence is essential to support the claims.

The introduction does not provide a sufficiently clear or logical rationale for specifically investigating GATA factors in the context of symbiotic nitrogen fixation. The transition from general transcription factor roles/GATA family functions to this specific focus needs strengthening. Please elaborate on the specific evidence or hypotheses motivating this targeted investigation.

Minor Concerns:

Redundancy in Results: The sentence in lines 123-125 is almost identical to the final sentence of the preceding paragraph. This redundant phrasing should be deleted.

Species Nomenclature: Latin binomial names for species are not italicized consistently throughout the text and figures, as required by standard biological nomenclature. This must be corrected everywhere.

Figure 2, The color scheme used for different branches/clades provides suboptimal contrast, making them difficult to distinguish. The colors of the branch lines are too similar to the colors of the terminal points, further hindering visual clarity and interpretation. Please revise the color palette to enhance contrast and distinction between branches and terminal points.

Figure 5, The figure appears overly schematic and lacks refinement. Many labels are positioned directly on lines or overlap graphical elements, making them hard to read and interpret. This figure requires significant improvement in layout, label positioning, and overall graphical quality for clarity.

Author Response

(The authors gave the same response as above.)

Reviewer 4 Report

Comments and Suggestions for Authors

The legend/description of figure 6 has the columns and rows reversed. 

As someone without a deep knowledge of GATA transcription factors, the categorizations used in your paper are confusing. To publish this paper, you need to inject a little more clarity into your categorizations and the relationship of the categories to each other. Initially, you describe four categories and seven subgroups in the introduction starting at line 40. This is based on a paper by Lowry et al, reference number 5. In your own analysis starting at line 105, you give a result of four categories with the same category names as those described in the introduction by Lowry et al. The paper by Lowry from your introduction is based on "zinc finger domains". Your own analysis starting on line 105 is based on, "conserved domain features and phylogenetic relationships." You state in your methods that, "the GATA zinc finger domain model was obtained from the Pfam database and utilized to screen the protein sequences.”. Presumably, you were trying to replicate the results of Lowry et al. by screening for zinc finger domains. Nevertheless, your results may not be identical to Lowry’s results. Did you check if your categories are fully overlapping with Lowry’s categories? 

Is the categorization shown at line 306 a completely new categorization based on a phylogenetic tree of the full length GATA protein, not just the zinc finger domain? Presumably, this is an entirely different categorization than the categorizations shown at lines 40 and 105 because it is a full-length sequence, or are these meant to be overlapping and describing the same categories at all three points in the paper? 

You state in lines 289-290 that class D is expressed in the roots. You state in lines 150-151 that class D diversified in angiosperms after separating from gymnosperms. You state in line 58 that class D is under strong selective pressure. When I read these earlier statements, I assumed that the conclusion of your paper would include a deeper analysis of this intriguing class of GATA transcription factors. Instead, class D is mysteriously dropped from the paper starting in section 2.6 of the paper and you never talk about this class again. You should at least tie up loose ends at the end of the paper and say something about this class, even if it was not the focus of your research.

Author Response

(The authors gave the same response as above.)

Round 2

Reviewer 2 Report

Comments and Suggestions for Authors

The authors have addressed most of the comments. The manuscript could be accepted after grammar checking.

Reviewer 3 Report

Comments and Suggestions for Authors

Overall, I am pleased to confirm that the authors have adequately addressed the majority of my previous concerns during the revision. The manuscript has been significantly improved and now presents a much stronger contribution to the field.